# Unified pre- and postsynaptic long-term plasticity enables reliable and flexible learning

**Rui Ponte Costa[1,2,3,4]\*, Robert C Froemke[5,6], P Jesper Sjöström[3†], Mark CW van Rossum[1†]**

[1]Institute for Adaptive and Neural Computation, School of Informatics, University of Edinburgh, Edinburgh, United Kingdom; [2]Neuroinformatics Doctoral Training Centre, School of Informatics, University of Edinburgh, Edinburgh, United Kingdom; [3]The Research Institute of the McGill University Health Centre, Department of Neurology and Neurosurgery, McGill University, Montreal, Canada; [4]Centre for Neural Circuits and Behaviour, University of Oxford, Oxford, United Kingdom; [5]Skirball Institute for Biomolecular Medicine, Departments of Otolaryngology, Neuroscience and Physiology, New York University School of Medicine, New York, United States; [6]Center for Neural Science, New York University, New York, United States

**Abstract** Although it is well known that long-term synaptic plasticity can be expressed both pre- and postsynaptically, the functional consequences of this arrangement have remained elusive. We show that spike-timing-dependent plasticity with both pre- and postsynaptic expression develops receptive fields with reduced variability and improved discriminability compared to postsynaptic plasticity alone. These long-term modifications in receptive field statistics match recent sensory perception experiments. Moreover, learning with this form of plasticity leaves a hidden postsynaptic memory trace that enables fast relearning of previously stored information, providing a cellular substrate for memory savings. Our results reveal essential roles for presynaptic plasticity that are missed when only postsynaptic expression of long-term plasticity is considered, and suggest an experience-dependent distribution of pre- and postsynaptic strength changes.

**\*For correspondence:** rui.costa@cncb.ox.ac.uk

[†]These authors contributed equally to this work

**Competing interests:** The authors declare that no competing interests exist.

Survival depends on learning accurate actions in response to sensory stimuli while remaining capable to quickly adapt in dynamic environments. The neural substrate of learning is believed to be long-term synaptic plasticity (*Pawlak et al., 2013*; *Nabavi et al., 2014*). After decades of debate (*MacDougall and Fine, 2013*; *Padamsey and Emptage, 2014*), it has become increasingly clear that expression of long-term synaptic plasticity can be either pre- or postsynaptic or both (*Zakharenko et al., 2001*; *Bayazitov et al., 2007*; *Sjöström et al., 2007*; *Loebel et al., 2013*; *Yang and Calakos, 2013*). However, the functional consequences of this segregation into pre- and postsynaptically expressed plasticity have remained unclear. To investigate this, we developed a biologically tuned spike-timing-dependent plasticity (STDP) model, that in contrast to earlier models (*Gerstner et al., 1996*; *Song et al., 2000*; *Senn et al., 2001*; *Seung, 2003*; *Froemke et al., 2006*; *Pfister and Gerstner, 2006*; *Leibold and Bendels, 2009*; *Clopath et al., 2010*; *Carvalho and Buonomano, 2011*; *Graupner and Brunel, 2012*; *Albers et al., 2013*), involves both loci of expression.

Inspired by earlier work (*Song et al., 2000*; *Pfister and Gerstner, 2006*), this phenomenological model relies on exponentially decaying traces of the pre- and postsynaptic spike trains, $X$ and $Y$ (*Figure 1A,B*). The presynaptic trace $x_+$ tracks past presynaptic activity, for example, glutamate

**eLife digest** Throughout life, animals must learn how to respond accurately to new sensations and environments, while retaining knowledge of previous experiences. Learning is widely believed to modify the connections (called synapses) between neurons of the cerebral cortex and other brain areas. This process is known as synaptic plasticity. Experimentally, presynaptic and postsynaptic changes have been identified, but it is not known what advantages there are to changing both components when, in principle, changing either might suffice.

To investigate this, Costa et al. developed a mathematical model of synaptic plasticity that captured both pre- and postsynaptic changes, based on a number of experiments over the last decade from recordings in the rat sensory cortices.

There were two major findings from this model. First, if both presynaptic and postsynaptic changes occur, the modeling results indicated that sensory perception could become more precise, as has been recently found in the rat auditory system. Second, because the details of presynaptic and postsynaptic changes are different, previously triggered changes leave behind a type of memory trace that allows apparently forgotten information to be rapidly relearned.

Interestingly, similar asymmetries have been reported in other brain regions. One future challenge is to understand whether these findings constitute a general principle of plasticity in the brain.

binding to postsynaptic NMDA receptors. When presynaptic activity $x_+$ is rapidly followed by post-synaptic spikes, unblocking NMDA receptors, postsynaptically expressed long-term potentiation (LTP) is triggered and increases the postsynaptic factor $q$, which can be interpreted as the quantal amplitude. Conversely, the postsynaptic trace $y_+$ represents prior postsynaptic activity, for example, retrograde nitric oxide (NO) signalling, which when paired with presynaptic spikes leads to presynaptically expressed LTP (*Sjöström et al., 2007*). Finally, the trace $y_-$ tracks postsynaptic activity such as endocannabinoid (eCB) retrograde release and elicits presynaptically expressed long-term depression (LTD) when coincident with presynaptic spikes (*Sjöström et al., 2003*). Presynaptically expressed plasticity is conveyed by long-term changes in the presynaptic factor $P$ (*Markram et al., 1998*), which can be interpreted as the presynaptic release probability (see 'Materials and methods').

The model parameters were tuned to an extensive data set of plasticity experiments of monosynaptic connections between neocortical layer-5 pyramidal cells (*Sjöström et al., 2001*, *2003*, *2007*). Homeostatic scaling of the postsynaptic amplitude $q$ was included to counterbalance postsynaptic potentiation (see 'Materials and methods') (*Turrigiano et al., 1998*). The resulting model not only captures the timing and frequency dependence of the synaptic strength changes (*Figure 1C* and *Figure 1—figure supplement 1*), but also its pre- as well as postsynaptic expression (*Figure 1D,E*). It thus captures the observed cross-scale interactions between short and long-term synaptic plasticity (*Sjöström et al., 2003*, *2007*). Short-term depression becomes weaker after LTD and stronger after LTP (*Figure 1F,G*). We validated the model against experiments with pharmacological blockade of presynaptic LTD or LTP (see 'Materials and methods'). Abolishing presynaptic LTP by NO blockade reduced total potentiation as only the postsynaptic potentiation component was left (*Sjöström et al., 2007*). Likewise, with the presynaptic trace $y_+$ disabled, presynaptic LTP was blocked, while the synaptic dynamics remained unchanged (*Figure 1H* and *Figure 1—figure supplement 3A*). Conversely, simulated blockade of presynaptic LTD during LTP induction gave rise to stronger presynaptic potentiation and short-term depression, as observed experimentally during eCB blockade (*Sjöström et al., 2007*) (*Figure 1H* and *Figure 1—figure supplement 3B*).

We first investigated the functional consequences of unified pre- and postsynaptically expressed STDP on the postsynaptic responses during cortical receptive field development. We simulated receptive field development of a postsynaptic neuron receiving 100 synaptic inputs ('Materials and methods'). Presynaptic activity was described by Poisson processes with rates spatially distributed according to a Gaussian profile (*Figure 2A*). We defined inputs near the peak of the Gaussian profile as *on*, and those far away from the peak as *off*. After learning, *on* neurons had increased $q$ and $P$,

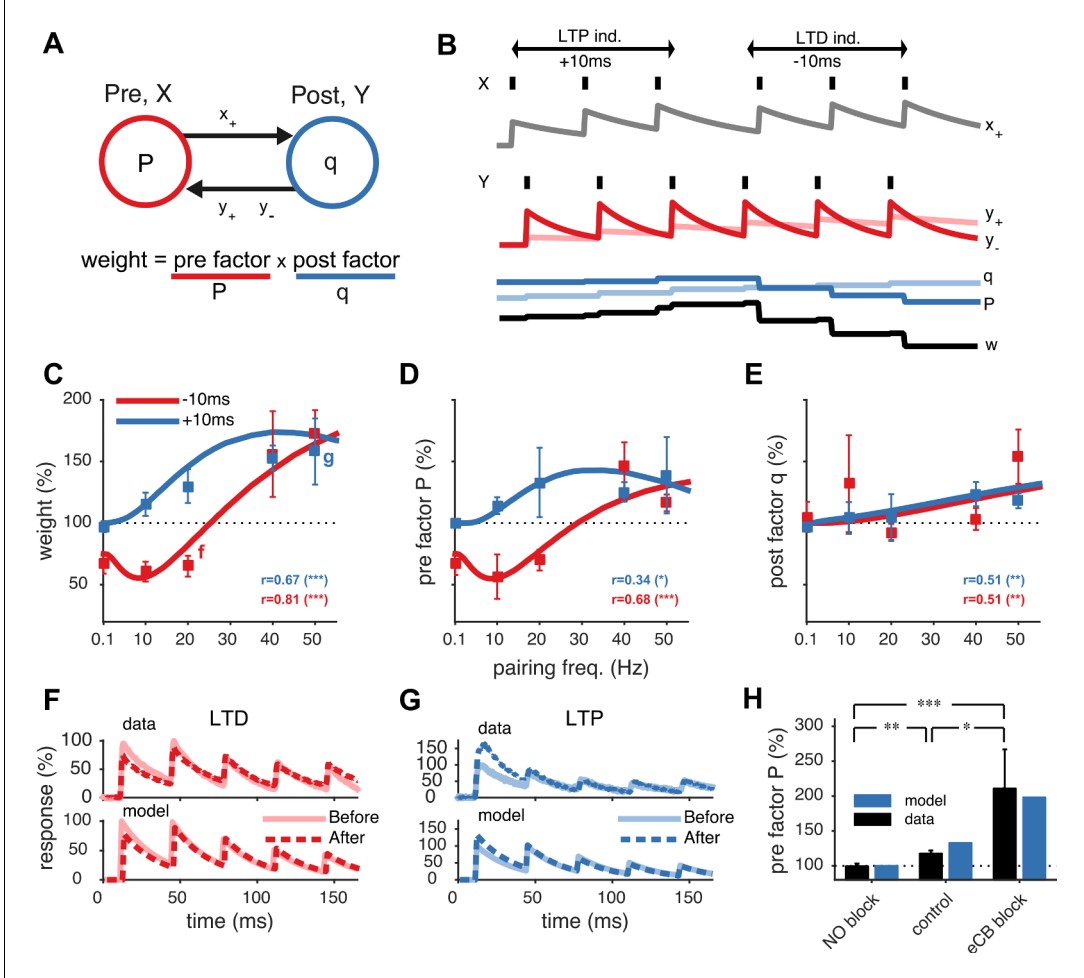

**Figure 1.** Unified model of pre- and postsynaptically expressed STDP. (**A**) The synaptic weight is the product of a presynaptic factor *P* and a postsynaptic factor *q*. Long-term modifications in *P* and *q* are governed by interactions between the pre- and postsynaptic spike trains. (**B**) Model example in which the postsynaptic neuron first spikes three times at 20 Hz (*Y*) $\Delta t$ = +10 ms after the presynaptic neuron (*X*), leading to LTP by increasing both *q* and *P*. Next, when the relative timing $\Delta t$ is reversed, long-term depression (LTD) results as *P* weakens strongly, even though *q* still slightly strengthens. (**C**) The model fits the rate dependence of synaptic plasticity (squares, (*Sjöström et al., 2001*)) for both positive (blue: +10 ms) and negative timings (red: −10 ms). (**D, E**) The changes in the pre- and postsynaptic factors *P* and *q* match experimental data (reanalyzed from *Sjöström et al., 2001*; see 'Materials and methods' and *Figure 1—figure supplement 2*). (**F, G**) As in experiments (top), short-term depression in the model is reduced after LTD (20 Hz, $\Delta t$ = −10 ms) and increased after LTP (50 Hz, $\Delta t$ = +10 ms) (bottom). Experimental traces from *Sjöström et al. (2003)* (**F**) and from *Sjöström et al. (2007)* (**G**). (**H**) Model (blue) is consistent with LTP experiments (black) (*Sjöström et al., 2007*) in control conditions, NO blockade, and eCB blockade. NO and eCB antagonists abolish and promote presynaptic LTP, respectively (*Sjöström et al., 2007*).

The following figure supplements are available for figure 1:

**Figure supplement 1.** The unified pre- and postsynaptic spike-timing-dependent plasticity (STDP) model (blue solid line) captured the characteristic temporal asymmetry of experimental STDP (black squares represent data from *Sjöström et al. (2001)*).

**Figure supplement 2.** Extraction of *P* and *q* from synaptic plasticity data from slice paired recordings using pharmacology and high frequency pairing (based on a long-step current injection plasticity protocol).

**Figure supplement 3.** Model is consistent with modifications of synaptic dynamics after pharmacological blockade of plasticity traces.

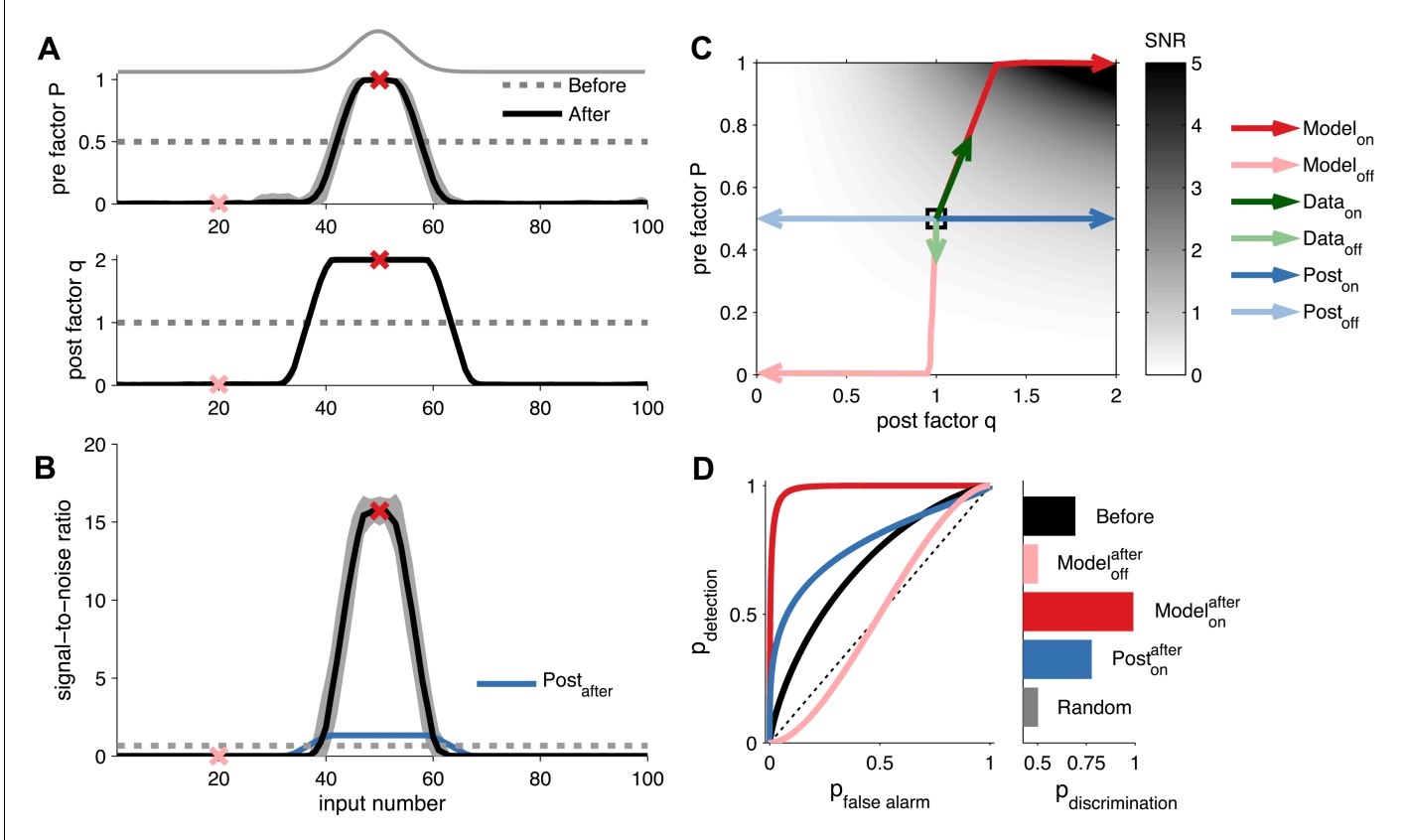

**Figure 2.** Unified pre- and postsynaptic plasticity improves receptive field discriminability. (**A**) Synaptic input rates follow a Gaussian spatial profile (solid grey line). Initially, the presynaptic factor *P* (top) and the postsynaptic factor *q* (bottom) are uniformly distributed (dashed lines). After learning, *P* (top) and *q* (bottom) both follow the input profile. Dark and light red crosses define examples of *on* and *off* receptive field positions, respectively. (**B**) After learning, the SNR is increased for *on* and decreased for *off* neurons. Postsynaptic plasticity alone leads to a smaller improvement (blue line). (**C**) While *on* neurons obtain higher SNR for postsynaptic-only potentiation (dark blue arrows), unified pre- and postsynaptic potentiation yields considerably better SNR (dark red arrows). *Off* neurons get lower SNR in both scenarios (light blue and light red arrows). Modifications of in vivo synaptic responses to a tone from *on* and *off* receptive field positions (dark and light green arrows, respectively; reanalyzed from ***Froemke et al. (2013)***, see 'Materials and methods') are consistent with unified pre- and postsynaptic expression but not with postsynaptic expression alone. The black square represents starting condition. Arrows represent the plasticity trajectory, where the model trajectories are plotted every 50 ms. (**D**) Only *on* positions with both pre- and postsynaptic plasticity yield near-perfect discrimination (dark red). Shown for comparison, the discrimination before development (black), after development for *off* neurons (light red), and after development for *on* neurons with postsynaptic expression only (blue).

The following figure supplements are available for figure 2:

**Figure supplement 1.** Long-term pre- and postsynaptic plasticity reduces response variability of receptive fields.

**Figure supplement 2.** Long-term pre- and postsynaptic plasticity improves signal-to-noise ratio (SNR) and information transmission in dynamic synapses.

**Figure supplement 3.** Extraction of effective *P* and *q* from in vivo receptive field plasticity experiments (data reanalyzed from ***Froemke et al. (2013)***.

while *off* neurons had reduced *q* and *P* (***Figure 2A***). During learning, the changes in *q* are preceded by changes in *P* (***Figure 2C***). To quantify the effect of the plasticity on the postsynaptic neuron, we stimulate a given input and calculated the signal-to-noise ratio (SNR) of the first postsynaptic response amidst background noise (see 'Materials and methods'). A high SNR means that the response can be easily distinguished from the background. After learning, only *on* inputs had

developed a high SNR (*Figure 2B*). Although both high and low *P* yielded low variance (*Figure 2—figure supplement 1*), high *P* was required for high SNR (*Figure 2C*).

To further assess the discriminability of the first postsynaptic response, we used classification analysis (see 'Materials and methods'), which revealed that *on* inputs obtained a near-perfect discrimination (*Figure 2D*) over a range of background noise levels (*Figure 2—figure supplement 1*). However, a model with only postsynaptic LTP, increasing *q* only, did not yield as reliable synaptic transmission (blue curve in *Figure 2C,D*)—maximal reliability required presynaptic LTP in addition. This is because, the variance of the first postsynaptic response increases quadratically with the postsynaptic factor *q* (see 'Materials and methods'). Our learning rule compensates for this increase in variance by also increasing the presynaptic factor *P*, thus making postsynaptic responses reliable and easier to discriminate. The increased discriminability does not only hold for the first response, but generalizes when considering the sum of the first *k* postsynaptic responses. Furthermore, the benefit of unified STDP remained when we compared the temporal information transmission across a range of presynaptic frequencies (*Figure 2—figure supplement 2*) (*Fuhrmann et al., 2002*; *Testa-Silva et al., 2014*).

The change in SNR and variability is consistent with recent sensory perception experiments (*Froemke et al., 2013*) in which pairing a tone with nucleus basalis stimulation led to an increased mean and a decreased variability of synaptic responses (*Figure 2—figure supplement 3*). Mapped to the parameters of the model, both *q* and *P* of the potentiated *on* responses increased (see 'Materials and methods'). Conversely, *off* responses that were depressed, decreased in *P* and did not significantly change in *q* (*Figure 2—figure supplement 3*), consistent with the initial modifications that the model predicts (*Figure 2C*). Therefore, unified pre- and postsynaptically expressed plasticity can account for the improved sensory perception after learning observed experimentally (*Froemke et al., 2013*). Furthermore our model suggests that both pre- and postsynaptic components should depend on sensory experience, in agreement with prior findings (*Finnerty et al., 1999*; *Cheetham et al., 2014*).

Plasticity should also allow the organism to quickly adapt to changing environments. Expression of layer-5 pyramidal cell STDP is curiously asymmetric: LTP is both pre- and postsynaptic (*Sjöström et al., 2007*), whereas LTD is expressed only presynaptically on the slice experiments timescale (*Sjöström et al., 2003*). In addition, presynaptic modifications are stronger than postsynaptic LTP (*Figure 1D,E*). To explore the consequences of this asymmetry, we extended the above network to study development when high rate inputs alternate between two locations. The neuron learned each receptive field by changes in the presynaptic factor *P* and the postsynaptic factor *q* (*Figure 3A–C*). When the stimulus location changed, however, the postsynaptic memory trace decayed only very slowly as a result of homeostatic scaling (*Figure 3B*). As a result, the neuron could rapidly relearn the previously acquired receptive field by just increasing *P*, which amounted to a 10-fold decrease in time to learn (*Figure 3D,E*). Unified pre- and postsynaptically expressed STDP thus allows for learning of new information while retaining hidden traces of prior experience.

Interestingly, spine changes in layer-5 pyramidal cells of visual cortex outlast sensory experience (*Hofer et al., 2008*), thus providing a structural substrate for the psychological phenomenon known as memory savings (*Ebbinghaus, 1913*). As synaptic structure and synaptic weight are closely correlated (*Matsuzaki et al., 2001*; *Holtmaat and Svoboda, 2009*), the memory savings mediated by structural spine plasticity (*Hofer et al., 2008*) are reminiscent of those provided by our unified plasticity model.

Here we have focused on neocortical data. Models based on synaptic traces are flexible and can describe both neocortical and hippocampal plasticity data (*Pfister and Gerstner, 2006*, and *Appendix 1*). We therefore expect that our modelling framework should also be able to capture plasticity in other brain regions, although with different parameters. For example, there are several key differences in the expression locus and in the speed of pre- and postsynaptic changes in hippocampus (*Bayazitov et al., 2007*). In cerebellum, there is evidence for the opposite asymmetry of expression, with LTP being pre- and postsynaptic, but LTD only postsynaptic (*Wang and Linden, 2000*; *Lev-Ram et al., 2003*).

In our work, memory savings are a consequence of the postsynaptic weight decay occurring on a much slower timescale than the presynaptic modifications. This arrangement, however, is not crucial for the predicted rapid relearning. What is necessary is that the synaptic strength is the product of pre- and postsynaptic components ($w = Pq$) and that these components evolve on different

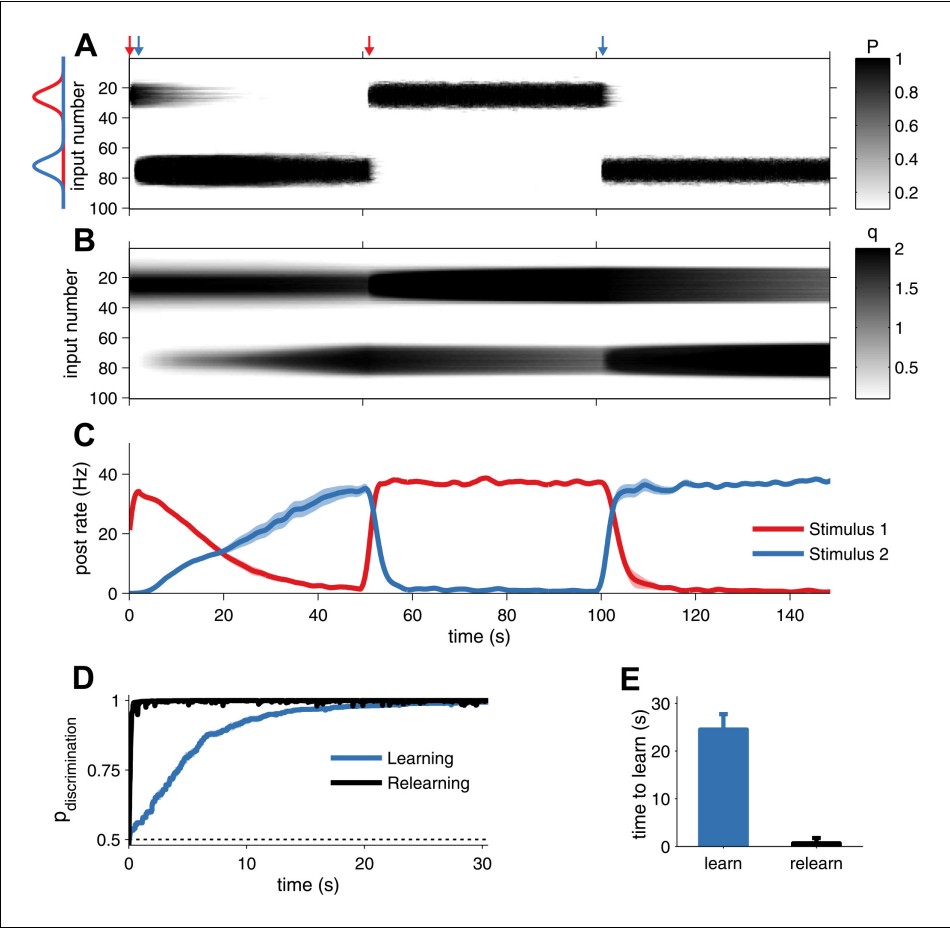

**Figure 3.** Unified pre- and postsynaptic STDP displays rapid relearning of previously experienced stimuli. (**A**) The presynaptic factor *P* follows the switching between two stimuli (red and blue profiles, arrows indicate switching time-points). (**B**) The postsynaptic factor *q*, however, is not erased and a trace of previously learned information remains, which decays slowly only due to synaptic homeostasis. The neuron was initially tuned to the red stimulus. The initial learning of the blue stimulus (at 1 s) was slow, but much faster the second time (at 101 s). (**C**) The neuron's tuning follows the two stimuli, as indicated by the alternating stimulus-specific spiking. Previously experienced stimuli are forgotten by the postsynaptic neuron, but a hidden trace remains. (**D**) Relearning occurs faster than learning. (**E**) Relearning was an order of magnitude faster than initial learning (time to reach 99% performance).

timescales. For example, fast postsynaptic changes combined with slow presynaptic changes would allow for the corresponding presynaptic trace of previous experience, which indeed could be the case in the cerebellum (*Wang and Linden, 2000*; *Lev-Ram et al., 2003*). Taken together, these findings suggest that plasticity expression asymmetry is not particular to neocortical layer-5 pyramidal cells, but rather a general functional principle that extends across different brain regions. Interestingly, similar functions can also be performed by neuronal inhibition, such as sharpening receptive fields (*Wilson et al., 2012*), keeping hidden memories in recurrent neural networks (*Vogels et al., 2011*), and acting as a substrate for memory savings in the cerebellum (*Medina et al., 2001*).

The existence of both pre- and postsynaptic expression of long-term synaptic plasticity has been enigmatic. Although it has been known that changes in release probability play a key role in determining the transmission of information across synapses (*Otmakhov et al., 1993*; *Stevens and Wang, 1994*; *Carvalho and Buonomano, 2011*), our theoretical treatment is the first to show that combined pre- and postsynaptic expression of long-term synaptic plasticity provides the brain with reliable sensory detection and the ability to quickly relearn previously experienced stimuli.

## Materials and methods

### Short- and long-term synaptic plasticity model

Short-term plasticity model

To model short-term synaptic plasticity, we used the Tsodyks-Markram model with facilitation (*Markram et al., 1998*). This model is defined by the following ODEs

$$\frac{dr(t)}{dt} = \frac{1 - r(t)}{D} - p(t)r(t)X(t), \tag{1}$$

$$\frac{dp(t)}{dt} = \frac{P - p(t)}{F} + P[1 - p(t)]X(t). \tag{2}$$

The first equation models the vesicle depletion process, where the (normalized) number of vesicles $r$ is decreased by an amount $p(t)r(t)$ after a presynaptic spike from the train $X(t) = \sum_{t_{pre}} \delta(t - t_{pre})$. Between spikes $r$ recovers to 1 with a depression time constant $D$. The second equation models the dynamics of the presynaptic factor $p$ which increases an amount $P[1 - p]$ after every presynaptic spike, decaying back to baseline presynaptic factor $P$ with a facilitation time constant $F$. By varying the synaptic dynamics parameters $D$, $F$ and $P$, one can obtain different synaptic dynamics. We used typical values for pyramidal-onto-pyramidal synapses (*Costa et al., 2013*), $D = 200$ ms and $F = 50$ ms, while $P$ is modified by long-term plasticity as below. The average number of vesicles released per spike is $r(t)p(t)$, which can be interpreted as the presynaptic strength.

### Long-term plasticity model

In layer-5 pyramidal to pyramidal cell synapses, timing-dependent LTD is presynaptically expressed. It is mediated by the coincidence between a postsynaptic signal (eCB release) and a presynaptic signal (presynaptic NMDA receptor activation) (*Sjöström et al., 2003, 2004*; *Bender and Feldman, 2006*; *Yang and Calakos, 2013*). LTP is driven by postsynaptic coincidence detection of the combined binding of glutamate and postsynaptic depolarization (*Bender and Feldman, 2006*; *Sjöström et al., 2007*; *Shouval et al., 2010*), promoting an increase in the number and/or properties of postsynaptic AMPA receptors (*Malinow and Malenka, 2002*). However, timing-dependent LTP also has a presynaptic component, mediated by postsynaptic diffusion of NO (*Hardingham and Fox, 2006*; *Sjöström et al., 2007*; *Hardingham et al., 2013*; *Yang and Calakos, 2013*).

Our phenomenological triplet model of long-term modification of pre- and postsynaptic components has three synaptic traces, two postsynaptic ($y_+$ and $y_-$) and one presynaptic ($x_+$), which increase upon a post- or presynaptic spike, respectively (see *Appendix 1* for a more detailed comparison with the triplet model (*Pfister and Gerstner, 2006*)). The traces are obtained by filtering the spike trains with a first-order low-pass filter. We defined the postsynaptic depression trace

$$\frac{dy_-(t)}{dt} = \frac{-y_-(t)}{\tau_{y_-}} + Y(t), \tag{3}$$

the postsynaptic potentiation trace

$$\frac{dy_+(t)}{dt} = \frac{-y_+(t)}{\tau_{y_+}} + Y(t), \tag{4}$$

and the presynaptic potentiation trace

$$\frac{dx_+(t)}{dt} = \frac{-x_+(t)}{\tau_{x_+}} + X(t). \tag{5}$$

The long-term modification in the weight is achieved by modifying the postsynaptic factor $q$ and the presynaptic factor $P$. The postsynaptic factor is modified with every postsynaptic spike $Y$ according to

$$\Delta q = c_+ \underbrace{x_+(t)y_-(t - \epsilon)Y(t)}_{\text{Triplet}_{\text{post}}^{\text{LTP}}}, \tag{6}$$

where $c_+$ is a constant that sets the amount of postsynaptic LTP. The $y_-$ trace is evaluated at $(t - \epsilon)$,

so that the value of the respective synaptic trace is readout before being updated. The triplet character of this rule is expressed by the fact that it contains the presynaptic component once, but the postsynaptic activity twice ($Y$ and filtered version $y_-$). This ensures that LTP only takes place when the postsynaptic spike follows both a presynaptic spike and a preceding postsynaptic spike (*Pfister and Gerstner, 2006*). As a result, low pairing frequencies do not lead to LTP, as $y_-$ will have decayed, consistent with data (*Sjöström et al., 2001*).

Similarly, the presynaptic factor is modified whenever the presynaptic cell is active according to

$$\Delta P = -d_- \underbrace{y_-(t)y_+(t)X(t)}_{\text{Triplet}^{\text{LTD}}_{\text{pre}}} + d_+ \underbrace{x_+(t-\epsilon)y_+(t)X(t)}_{\text{Triplet}^{\text{LTP}}_{\text{pre}}}. \tag{7}$$

For plasticity in $P$ to occur, the presynaptic spikes $X$ readout the postsynaptic traces (presynaptic coincidence detection), $y_-y_+$ for presynaptic LTD and $x_+y_+$ for presynaptic LTP. $d_-$ and $d_+$ are constants that set the amount of presynaptic LTD and LTP, respectively. While presynaptic LTD has a triplet form, it contains two postsynaptic traces and the raw presynaptic spike train. Therefore it does not vanish at low frequencies. Equivalently, this term could be written as a doublet rule with a double exponential as the presynaptic trace.

The total synaptic strength is a product of both pre- and postsynaptic factors

$$w(t) = qp(t)r(t). \tag{8}$$

For a synapse that has not been stimulated recently this simplifies to $w = Pq$.

Being a probability we hard-bounded $P = [0, 1]$. The postsynaptic factor $q$ had a lower bound of 0, and an upper bound of 2. Alternatively a soft-bounded rule could be used (*van Rossum et al., 2012*). In the data used to fit the model (see below), postsynaptic homosynaptic LTD was not apparent on the timescale of the experiment. Because it seems unrealistic that the postsynaptic factor $q$ never decreases, slow homeostasic scaling of the postsynaptic factor was included for network simulations (*Turrigiano et al., 1998*). This prevents weakly active synapses from potentiating the postsynaptic factor $q$. It was modelled as a postsynaptic subtractive normalization, so that the total change in $q$ of synapse $i$ was equal to $\Delta q_i - \alpha \frac{1}{N}\sum_{j=1}^{N}\Delta q_j$ (*Miller and MacKay, 1994*). The only condition on the speed $\alpha$ for it to be consistent with the data, is that it should not lead to noticable homeostasis on the timescale of the experiments. For computational efficiency we used $\alpha = 0.075$, which is still orders of magnitude faster than what has been observed in homeostasis experiments. The exact form of slow normalization ($\alpha \to 0$) does not affect the qualitative behavior of the model. Note that the timescale of the slow normalization determines how long the memory savings effects are present.

To speed up the numerical implementations, we integrated the synaptic traces between the pre- and postsynaptic spikes. In the following equations, we label the presynaptic spikes with $k$ and the postsynaptic ones with $l$.

$$y_-^{l+1} = y_-^l \exp\left(-\frac{\Delta t_{\text{post}}}{\tau_{y_-}}\right) + 1, \tag{9}$$

$$y_+^{l+1} = y_+^l \exp\left(-\frac{\Delta t_{\text{post}}}{\tau_{y_+}}\right) + 1, \tag{10}$$

$$x_+^{k+1} = x_+^k \exp\left(-\frac{\Delta t_{\text{pre}}}{\tau_{x_+}}\right) + 1. \tag{11}$$

We subsequently integrated the model between pre- and postsynaptic spikes

$$q_{l+1} = q_l + c_+ x_+^k \exp\left(-\frac{\Delta t_{\text{post−pre}}}{\tau_{x_+}}\right) y_-^l \exp\left(-\frac{\Delta t_{\text{post}}}{\tau_{y_-}}\right), \tag{12}$$

$$P_{k+1} = P_k - d_- y_-^l \exp\left(-\frac{\Delta t_{\text{pre−post}}}{\tau_{y_-}}\right) y_+^l \exp\left(-\frac{\Delta t_{\text{pre−post}}}{\tau_{y_+}}\right) + d_+ y_+^l \exp\left(-\frac{\Delta t_{\text{pre−post}}}{\tau_{y_+}}\right) x_+^k \exp\left(-\frac{\Delta t_{\text{pre}}}{\tau_{x_+}}\right), \tag{13}$$

where $\Delta t_{\text{post−pre}}$ is the time between the current postsynaptic spike and the last presynaptic spike, $\Delta t_{\text{post}}$ is the time between the current postsynaptic spike and the last one, and similarly for $\Delta t_{\text{pre−post}}$

and $\Delta t_{\mathrm{pre}}$. Finally, we also integrated the STP (*Equations 1, 2*) between presynaptic spikes $k$ and $k + 1$, a time $\Delta t_{\mathrm{pre}}$ apart, yielding

$$r_{k+1} = 1 - [1 - r_k(1 - p_k)]\exp\left(-\frac{\Delta t_{\mathrm{pre}}}{D}\right), \tag{14}$$

$$p_{k+1} = P + p_k[1 - P]\exp\left(-\frac{\Delta t_{\mathrm{pre}}}{F}\right). \tag{15}$$

with initial conditions $r_0 = 1$ and $p_0 = P$.

## Model fitting to in vitro plasticity data

We fitted the free parameters of the long-term plasticity model $\theta = \{d_-, \tau_{y-}, d_+, \tau_{y+}, c_+, \tau_{x+}\}$ to the frequency- and timing-dependent slice STDP data of layer-5 pyramidal cells (*Sjöström et al., 2001*). Parameters are shown in *Table 1*. Rather than fitting to changes in the weight $w$, we fitted directly to modifications in $P$ and $q$ (see *Equations 21, 22* for our estimators of $P$ and $q$). This was done by minimizing the mean squared error between the data and the experiments for both $P$ and $q$ (as shown in *Figure 1*)

$$\theta = \mathrm{argmin}_\theta \frac{1}{N} \sum_j^N \left[ \left( \frac{P_{\mathrm{model}}^{\mathrm{after}}}{P_{\mathrm{model}}^{\mathrm{before}}} - \frac{P_{\mathrm{data}}^{\mathrm{after}}}{P_{\mathrm{data}}^{\mathrm{before}}} \right)^2 + \left( \frac{q_{\mathrm{model}}^{\mathrm{after}}}{q_{\mathrm{model}}^{\mathrm{before}}} - \frac{q_{\mathrm{data}}^{\mathrm{after}}}{q_{\mathrm{data}}^{\mathrm{before}}} \right)^2 \right], \tag{16}$$

where $N$ denotes the number of protocols fitted, 10 in total (5 different pairing frequencies with $-10$ ms or $+10$ ms relative timing, see below). For induction protocols at high frequencies ($\geq 10$ Hz), pre- and postsynaptic spike trains consisted of five spikes that were paired 15 times at 0.1 Hz. Low-frequency pairings (0.1 Hz) were done with a single pre- and postsynaptic spike (as in *Sjöström et al., 2001*). Before plasticity induction, $P$ and $q$ were set to 0.5 and 1, respectively. For the interaction of STP and STDP simulations (*Figure 1F,G*), we used a standard passive neuron model with a membrane time constant of 25 ms.

Without further fitting this model also captured pharmacological blockade of the plasticity traces. In the model, we simulated the experimental effects of pharmacological blockade by setting the relevant parameter or variable to 0. Specifically, we simulated the effects of blocking two different retrograde messenger systems shown to be involved in STDP in layer-5 pyramidal cell pairs, eCB signaling (*Sjöström et al., 2003*) and NO signaling (*Sjöström et al., 2007*). To reproduce pharmacological blockade experiments, we used high-frequency pairing (50 Hz) with $+10$ ms delay, which is comparable with our frequency-dependent results and approximates the long depolarizing currents used in *Sjöström et al. (2007)*. Blocking eCB receptors prevents presynaptic LTD (*Sjöström et al., 2003*). By setting $d_- = 0$ presynaptic LTD was disabled. This reveals presynaptic LTP and enhances short-term depression (*Figure 1—figure supplement 3*), consistent with experimental evidence (*Sjöström et al., 2007*), as the drugs used are likely to block presynaptic eCB receptors. In contrast, blocking NO decreases LTP but does not affect short-term synaptic dynamics (*Sjöström et al., 2007*) (*Figure 1—figure supplement 3A*). We simulated this by setting $y_+ = 0$, so that both presynaptic components were absent.

## Stochastic synaptic responses and in vitro *P* and *q* estimation

The release of neurotransmitter was assumed to follow a standard binomial model (*Del Castillo and Katz, 1954*)

**Table 1.** Unified pre- and postsynaptic spike-timing-dependent plasticity (STDP) model parameters

| Parameter | $d_-$ | $\tau_{y-}$ (ms) | $d_+$ | $\tau_{y+}$ (ms) | $c_+$ | $\tau_{x+}$ (ms) |
|---|---|---|---|---|---|---|
| Young rat visual cortex | 0.00389 | 12.5 | 0.002483 | 417.8 | 0.013706 | 48.6 |

The model was fitted to data from young rat visual cortex (*Sjöström et al., 2001*).

**Table 2.** Comparison between unified pre- and postsynaptic STDP model and different versions of the *triplet* model (for simplicity we removed the function arguments) (**Pfister and Gerstner, 2006**)

|  | LTD | LTP$_1$ | LTP$_2$ |
|---|---|---|---|
| pre-post STDP | $X\ d_- y_- y_+$ | $X\ d_+ y_+ x_+$ | $Y\ c_+ x_+ y_-$ |
| minimal HC Triplet | $X\ A_2^- y_1$ | $Y\ A_2^+ x_1$ | $Y\ A_3^+ x_1 y_2$ |
| minimal VC Triplet | $X\ A_2^- y_1$ | – | $Y\ A_3^+ x_1 y_2$ |

$$P_{\text{syn}}(X=k) = \binom{N}{k} P^k (1-P)^{N-k}, \tag{17}$$

which defines the probability of having *k* successful events (neurotransmitter release) given *N* trials (release sites) with equal probability *P*.

The mean synaptic response is scaled by a postsynaptic factor *q*, which can be related to the quantal amplitude so that

$$\mu_{\text{syn}} = PqN, \tag{18}$$

and the variance is

$$\sigma_{\text{syn}}^2 = q^2 NP(1-P). \tag{19}$$

Following the binomial release model (**Equation 18**), $\mu_{\text{syn}}$ (**Equation 19**) and $\sigma_{\text{syn}}^2$ (**Equation 20**),

$$P = \frac{\mu_{\text{syn}}}{Nq}, \tag{20}$$

and

$$q = \frac{\sigma_{\text{syn}}^2}{\mu_{\text{syn}}} + \frac{\mu_{\text{syn}}}{N}. \tag{21}$$

The number of release sites *N* is believed to change only after a few hours (**Bolshakov et al., 1997**; **Saez and Friedlander, 2009**). As the slice synaptic plasticity experiments analysed here lasted only up to 1.5 hr (**Sjöström et al., 2001**) and we were interested in the relative changes we assumed constant $N = 5.5$ in our analysis below, as estimated in **Markram et al. (1997)** using data from the same connection type we used to fit our model. **Equations 21, 22** were used to estimate *P* and *q* from in vitro plasticity data (see above), respectively (dataset deposited at Dryad data repository at 10.5061/dryad.p286g [**Costa et al., 2015**]). Note that because the data had to be reanalized in full there are minor differences in the mean weights previously published (**Sjöström et al., 2001**).

We verified our *P* and *q* extraction method by analysing short-term plasticity experiments with pharmacological manipulation of presynaptic release or of postsynaptic gain (**Figure 1—figure supplement 2A**, **Sjöström et al., 2003**), and experiments with pharmacological blockade of pre- or postsynaptic long-term plasticity (**Figure 1—figure supplement 2B**, **Sjöström et al., 2007**) (**Figure 1—figure supplement 2A,B**). In addition, long-term changes in *P* but not in *q* were inversely correlated with changes in paired-pulse ratio, as expected (**Figure 1—figure supplement 2C,D**). Taken together, these results lend experimental support to our binomial-distribution-based approach for extracting *P* and *q* to tune changes in the pre- and postsynaptic modifications of our unified STDP model (**Figure 1D,E**).

## Analysis of in vivo data

We extracted the effective *P* and *q* from the in vivo data obtained by **Froemke et al. (2013)**. Again using a binomial model, we obtained estimators for their variability measure given by $v = q\,(1 - P)$ and the mean by $\mu = PqN$. To ease comparison with our simulations we set the initial *P* to the same initial condition used in our simulations $P = 0.5$ (**Costa et al., 2013**). We then obtained the initial $N = \frac{|\mu|}{qP}$ and the initial $q = \frac{v}{(1-P)}$. For the after pairing data we allowed both pre- and postsynaptic factors *P* and *q* to change, while *N* was fixed to the values extracted before pairing (**Bolshakov et al.,**

1997). The estimations after learning were obtained as $q = v + \frac{|\mu|}{N}$ and $P = \frac{|\mu|}{Nq}$. We used these estimators to extract $q$ and $P$ from measurements for both the depression experienced for the unpaired (best before pairing) receptive field position and the potentiated paired position (*Froemke et al., 2013*). After pairing, the effective $q$ of the potentiated ('on') response increased from $q_{\text{before}}^{\text{on}} = 23.3$ pA to $q_{\text{after}}^{\text{on}} = 27.1$ pA (+16.3%), while $P$ increased from $P_{\text{before}}^{\text{on}} = 0.5$ to $P_{\text{after}}^{\text{on}} = 0.73$ (+46%). Responses that were depressed ('off'), typically the original best frequency, yielded no statistically significant change in $q_{\text{before}}^{\text{off}}$, while $P_{\text{before}}^{\text{off}} = 0.5$ and $P_{\text{after}}^{\text{off}} = 0.40$ (−20%) (*Figures 2*, *Figure 2—figure supplement 1* and *Figure 2—figure supplement 3*). To ease comparison with the postsynaptic factor in the simulations we scaled the experimentally obtained $q$ such that before plasticity it was 1. We compared models where we allowed both $P$ and $q$ to change or only one of them, the lower variability estimation error was obtained by the one where both factors change (*Figure 2—figure supplement 3E*). The estimation error was calculated as $\frac{1}{N}\sum_i^N (v_{\text{real}}^i - v_{\text{estimated}}^i)^2$, where $N$ is the number of data points.

## Synaptic signal detection

We calculated the SNR of a synaptic response defined here by a random variable $s$, amidst additive background noise defined by the random variable $n$ as follows

$$\text{SNR}_{\text{syn}} = 2\frac{(\langle s \rangle - \langle n \rangle)^2}{\sigma_s^2 + \sigma_n^2}, \tag{22}$$

It is assumed that $n \sim \mathcal{N}(0, \sigma_n^2)$ and we also used the Gaussian approximation to the binomial release model specified above, $s \sim \mathcal{N}\left(PqN, q^2NP(1-P) + \sigma_n^2\right)$, from which follows the SNR of the first postsynaptic response

$$\text{SNR}_{\text{syn}} = 2\frac{(PqN)^2}{q^2NP(1-P) + 2\sigma_n^2}. \tag{23}$$

In *Figure 2*, we used $\sigma_n^2 = 0.5$. Variance of the $k$-th postsynaptic response is given by $\sigma_{\text{syn}^k}^2 = q^2Nr_kp_k(1 - r_kp_k)$ (*Figure 2—figure supplement 2A*). The SNR of the $k$-th postsynaptic response is

$$\text{SNR}_{\text{syn}}^k = 2\frac{(r_kp_kqN)^2}{q^2Nr_kp_k(1 - r_kp_k) + 2\sigma_n^2}, \tag{24}$$

where $p_k$ and $r_k$ are given by *Equations 15, 16*, respectively. The SNR of the sum of the first $K$ responses, evoked at a given presynaptic firing rate $\rho$ therefore equals

$$\text{SNR}_{\text{syn}}^\rho = 2\frac{\left(\sum_{k=0}^{K-1} r_kp_kqN\right)^2}{\sum_{k=0}^{K-1} q^2Nr_kp_k(1 - r_kp_k) + 2\sum_{k=0}^{K-1}\sigma_n^2}. \tag{25}$$

After unified STDP the first response has a higher amplitude and the second one a much lower amplitude due to synaptic depression. Combined with the background noise, the SNR can drop when the second or further responses are included. However, the SNR of the summed response will always be larger than when only postsynaptic modifications are made (see *Figure 2—figure supplement 2B*). This holds for any frequency, *Figure 2—figure supplement 2C* and carries over to an information theoretic analysis of the response, *Figure 2—figure supplement 2D*.

Next, we used ROC analysis to compute the *false alarm* and *detection* probability of the first postsynaptic response

$$p_{\text{false alarm}} = \int_T^{+\infty} P_n(r)dr = \frac{1}{2}\text{erfc}\left(\frac{T}{\sqrt{2\sigma_n^2}}\right), \tag{26}$$

$$p_{\text{detection}} = \int_T^{+\infty} P_s(r)dr = \frac{1}{2}\text{erfc}\left(\frac{T - PqN}{\sqrt{2q^2NP(1-P) + \sigma_n^2}}\right). \tag{27}$$

where $T$ is the discrimination threshold, and erfc is the complementary error function defined as $\text{erfc}(x) = \frac{2}{\sqrt{\pi}} \int_x^\infty e^{-t^2} dt$. To assess the overall discriminability, we used $p_{\text{discrimination}}$, which is the area under the ROC curve (AUC). The AUC was computed by integrating over the ROC curve using the trapezoid method (see *Figure 2D*). Given that $N$ is a simple constant we set it to 1, unless otherwise stated (see data inference above).

## Receptive field development

For the receptive field development simulations, we used a feedforward network with 100 presynaptic neurons $j$ with Poisson statistics and a single integrate-and-fire postsynaptic neuron. The postsynaptic neuron was modelled as an adaptive exponential integrate-and-fire neuron model (*Brette and Gerstner, 2005*). Model parameters were as reported in *Brette and Gerstner (2005)*; *Badel et al. (2008)* and synapses were modelled as input currents. The firing rate of the presynaptic Poisson neurons was modelled using a Gaussian profile, defined as

$$\rho(j; p, \sigma) = \rho_{\min} + (\rho_{\max} - \rho_{\min}) e^{\frac{-(j-p)^2}{2\sigma^2}}. \tag{28}$$

where $\rho$ is the rate in the Poisson neuron model $j$, $p$ the input position for which the rate is maximal, and $\sigma$ = 5 Hz the distribution spread. $\rho_{max}$ and $\rho_{min}$ are the maximum and minimum rates, and were set to $\rho_{\max}$ = 50 Hz and $\rho_{\min}$ = 3 Hz. We scaled $d_-$, $d_+$ and $c_+$ by a factor 0.15 to yield a smoother receptive field development. $q$ was bounded between 0 nA and 20 nA, so that the synaptic input is appropriately scaled for the neuron model used. The network was simulated for 100 s to achieve convergence. For the memory savings experiment, we interleaved two receptive field positions. Results for receptive development and memory savings were averaged over 10 runs. The response of the postsynaptic neuron (*Figure 3C*) was assessed by presenting each stimulus alone with long-term synaptic plasticity inactive. Receptive field simulations were implemented in simulator Brian 1.41 (*Goodman and Brette, 2008*). Code for running and plotting the savings experiment is available online (http://modeldb.yale.edu/184487).

## Statistical comparison

Results are reported as mean ± SEM. Statistical comparisons were made with Student's *t*-test for equal means, if data was normally distributed as assessed using Kolmogorov–Smirnov test, Mann–Whitney U non-parametric test was used otherwise. For multiple comparisons we applied ANOVA or Kruskal–Wallis test for normally or non-normally distributed data, respectively. For correlation analysis the Spearman's coefficient was used together with one-tailed Student's *t*-test. Significance levels are *$p < 0.05$, **$p < 0.01$, and ***$p < 0.001$.

## Acknowledgements

We thank Gary Bhumbra, Megan Carey, Claudia Clopath, Wulfram Gerstner, Matthias Hennig, Robert Legenstein, Wolfgang Maass, Miha Pelko, Jean-Pascal Pfister, Paolo Puggioni, João Sacramento, Walter Senn, Lukas Solanka, the Sjöström lab, the Vogels lab and the Center for Neural Circuits and Behaviour for useful discussions. We thank Michelle Giugliano for sharing his code to calculate synaptic information transmission with dynamic synapses.

## Additional information

### Author contributions

RPC, Conception and design, Analysis and interpretation of data, Drafting or revising the article; RCF, Analysis and interpretation of data, Drafting or revising the article; PJS, MCWvR, Co-senior author, Conception and design, Analysis and interpretation of data, Drafting or revising the article

## Author ORCIDs

Rui Ponte Costa, http://orcid.org/0000-0003-2595-2027
P Jesper Sjöström, http://orcid.org/0000-0001-7085-2223
Mark CW van Rossum, http://orcid.org/0000-0001-6525-6814

## Funding

| Funder | Grant reference number | Author |
|---|---|---|
| Engineering and Physical Sciences Research Council | EP/F500385/1 | Rui Ponte Costa |
| Medical Research Council | | Rui Ponte Costa |
| Biotechnology and Biological Sciences Research Council | BB/F529254/1 | Rui Ponte Costa |
| Fundação para a Ciência e a Tecnologia | SFRH/BD/60301/2009 | Rui Ponte Costa |
| National Institute on Deafness and Other Communication Disorders | DC009635 | Robert C Froemke |
| Albert Einstein College of Medicine of Yeshiva University | Hirschl/Weill-Caulier Career Research Award | Robert C Froemke |
| Alfred P. Sloan Foundation | Sloan Research Fellowship | Robert C Froemke |
| National Institute on Deafness and Other Communication Disorders | DC012557 | Robert C Froemke |
| European Commission | 243914 | P Jesper Sjöström |
| Canada Foundation for Innovation | 28331 | P Jesper Sjöström |
| Canadian Institutes of Health Research | 126137 | P Jesper Sjöström |
| Natural Sciences and Engineering Research Council of Canada | 418546-2 | P Jesper Sjöström |

The funders had no role in study design, data collection and interpretation, or the decision to submit the work for publication.

## Additional files

### Major datasets

The following dataset was generated:

| Author(s) | Year | Dataset title | Dataset URL | Database, license, and accessibility information |
|---|---|---|---|---|
| Costa RP, Froemke RC, Sjöström PJ, van Rossum MCW | 2015 | Data from: Unified pre- and postsynaptic long-term plasticity enables reliable and flexible learning | http://dx.doi.org/10.5061/dryad.p286g | Available at Dryad Digital Repository under a CC0 Public Domain Dedication |

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

## Appendix 1

### Comparison between unified pre- and postsynaptic STDP model, and triplet STDP model (*Pfister and Gerstner, 2006*)

Our model has some similarities with the triplet-STDP model introduced in *Pfister and Gerstner (2006)*, however note that the triplet model does not distinguish between pre- and postsynaptic components of expression. The triplet model is defined by the following components: presynaptic traces, $x_1$ and $x_2$, and postsynaptic traces $y_1$ and $y_2$. The weight changes are modelled as a combination of pair and triplet components (*full Triplet* model) as follows

$$\Delta w^- = -X(t)y_1[A_2^- + A_3^- \ x_2(t-\epsilon)], \tag{29}$$

$$\Delta w^+ = Y(t)x_1[A_2^+ + A_3^+ \ y_2(t-\epsilon)]. \tag{30}$$

However, to fit the intra-pairing frequency observed in the young rat visual cortex (VC) (*Sjöström et al., 2001*), a reduced model ($A_3^- = 0$ and $A_2^+ = 0$) was found to be sufficient (*minimal VC Triplet*) (*Pfister and Gerstner, 2006*)

$$\Delta w^- = -X(t)A_2^- \ y_1, \tag{31}$$

$$\Delta w^+ = Y(t)A_3^+ \ x_1y_2(t-\epsilon). \tag{32}$$

Moreover, another slightly more complex model ($A_3^- = 0$) was found to be able to capture triplet and quadruplet experiments performed in the hippocampus (HC) (*Wang et al., 2005*) (*minimal HC Triplet*)

$$\Delta w^- = -X(t)A_2^- y_1, \tag{33}$$

$$\Delta w^+ = Y(t)x_1[A_2^+ + A_3^+ y_2(t-\epsilon)]. \tag{34}$$

Interestingly, our model also has two LTP and one LTD components, that can be mapped onto the *minimal HC Triplet* (see *Table 2*). However, to capture the pharmacological blockade experiments reported in *Sjöström et al. (2007)*, we needed three triplets, rather than one triplet and two doublets as in the *minimal HC Triplet* model.

