## [Decision Letter]

Thank you for submitting your work entitled “Unified pre- and postsynaptic long-term plasticity enables reliable and flexible learning” for peer review at *eLife*. Your submission has been favorably evaluated by Eve Marder (Senior Editor) and three reviewers, one of whom is a member of our Board of Reviewing Editors.

The reviewers have discussed the reviews with one another and the Reviewing Editor has drafted this decision to help you prepare a revised submission.

All three reviewers found aspects of the improvement in SNR confusing. I think this can be remedied with textual changes not requiring additional experiments or simulations (although if additional simulations – e.g. addressing Reviewer #3's concerns about frequency – will help to clarify, these could presumably be included). All three reviews are included below in the hopes that this will aid in clarifying the manuscript.

*Reviewer #1:*

This is a carefully done, compact paper with a crisp message that offers a potential solution to an important problem in neuroscience. Although there is now broad agreement that pre- and postsynaptic plasticity mechanisms can coexist at many central synapses, the functional benefit of this is unclear. The authors here adapt prior models of spike-timing-dependent plasticity to take account of the experimentally determined properties of pre- and postsynaptic expression mechanisms for plasticity. They show two functional benefits. First, the two mechanisms better account for observed improvements in sensory discrimination with learning and second, the dual mechanism permits much more rapid relearning when the stimuli being learned change with time.

I have no major concerns.

*Reviewer #2:*

It is really good to see a model of this detail and rigor combining the pre and post-synaptic aspects of plasticity and to see that we have finally arrived at a point where it is accepted that pre and post-synaptic elements show plasticity. My comments are going to be fairly minor but I hope that the authors will make some effort to tackle them because it will probably help communicate the ideas.

The first point is that the model seeks to explain the advantages of presynaptic plasticity. One of these is the decrease in variance in developing receptive fields another is retained information or savings on reversal of plasticity. Unfortunately, having raked through the formulae a few times, I cannot see where this comes from. In Figure 3, the relearning occurs more quickly because the post-synaptic factor decays slowly. Can the authors explain in more detail why presynaptic plasticity improves relearning? Also, could it not (theoretically) be achieved another way, say by increasing the duration of the post-synaptic factor? I am still missing what is unique about the disposition of the presynaptic plasticity.

I am not disputing the premise, however. It may well be that the slower turnover of presynaptic terminals compared to spines allows faster relearning. I believe that Finnerty and colleagues have done some work on this and they should probably be mentioned in this paper.

The second point is that the main body of the paper seems to be over very quickly. I was left searching for more text. There are several points that could be more fully discussed, including the elements that are explicitly missing from the model and what the impact might be. For example, how does inhibition and plasticity of inhibition impact the outcome and how might the time course of structural plasticity modify the model. On the second point, the Matsuzaki reference deals with structure and synaptic weight post-synaptically but not pre-synaptically. It would also be useful to hear how the model might fit different structures - would more presynaptic plasticity be warranted in layer 5 pyramidal cells than in hippocampus for example?

*Reviewer #3:*

This paper presents a theoretical model for STDP that incorporates both a pre- and post-synaptic expression locus. To date the vast majority of models have focused on implementations of LTP and LTD that simply scale the value of the synaptic weight, but do not alter short-term synaptic plasticity-thus altering the temporal profile of EPSPs. Importantly the model is based on, and captures, the experimental data which strongly suggests that there are pre- and postsynaptic expression mechanisms. Together the paper is highly novel and provides important data-based insights to our understanding of synaptic plasticity.

It is very interesting that the rule results in better discrimination. What is not clear is why? That is, why does a higher P value provide improve SNR? Yes, the first presynaptic spike generates a large reliable EPSP, but the subsequent spikes are less reliable and depressed. The SNR and ROC analyses were analytical, but I'm struggling to understand how these analyses can capture the true SNR and discriminability without taking into account the firing rate of the presynaptic units. Discriminability must depend in part of on the frequency of the Poisson inputs (and the D and F time constants)-but as it stands the analysis is based on the probabilistic nature of P, that is, essentially only for the first spike. Either I'm missing something, or an analysis based on the actual simulations (which take into account the timing of the spikes) should be performed. This is probably a minor issue, however, as Figure 3 makes it clear the discriminability based on firing rate is excellent.

Based on Figure 3, P seems to saturate at 1, meaning that there is little or no release for the next 200 ms. This is one reason the SNR calculation may be misleading. It would be useful to show postsynaptic trace segments during stimulation as part of this figure.

I think some readers, particularly, those accustomed to the hippocampal field, will be unfamiliar with the strong evidence that there are presynaptic components to STDP. So I think it is important to strengthen their argument a bit, and perhaps mention that, these studies are based on neocortical data, and that there is likely a difference between hippocampal and neocortical STDP.

---

## [Author Response]

*All three reviewers found aspects of the improvement in SNR confusing. I think this can be remedied with textual changes not requiring additional experiments or simulations (although if additional simulations – e.g. addressing Reviewer #3's concerns about frequency – will help to clarify, these could presumably be included). All three reviews are included below in the hopes that this will aid in clarifying the manuscript*.

These are important points. We have modified the manuscript considerably to address this. First, we have now clarified how the SNR is calculated. Briefly, it is the ratio between the synaptic signal (i.e. the first EPSP) and background noise. We have also further clarified why presynaptic plasticity is important in improving the SNR.

Additionally, in response to Reviewer 3, we have extended our SNR analysis for trains of input. Finally, we also calculated the information transmission at different frequencies for dynamics synapses, as proposed by [17]. This is part of the new Figure 2—figure supplement 3, which shows that SNR and information transmission increases across different frequencies, when both pre- and postsynaptic plasticity occurs, but not when only postsynaptic plasticity occurs.

Reviewer #2:

*The first point is that the model seeks to explain the advantages of presynaptic plasticity. One of these is the decrease in variance in developing receptive fields another is retained information or savings on reversal of plasticity. Unfortunately, having raked through the formulae a few times, I cannot see where this comes from*.

We have tried to improve the intuition behind both sets of results. For the change in variance this is now clarified. For the savings result, we clarified the explanation of our results.

*In*Figure 3*, the relearning occurs more quickly because the post-synaptic factor decays slowly. Can the authors explain in more detail why presynaptic plasticity improves relearning? Also, could it not (theoretically) be achieved another way, say by increasing the duration of the post-synaptic factor? I am still missing what is unique about the disposition of the presynaptic plasticity*.

Yes, the rapid relearning occurs because the postsynaptic weight decay is slower than the presynaptic modifications. In theory this could be achieved by other mechanisms; however, our model was in fact inspired by the experimental observation that the presynaptic modifications are bidirectional (expressing both LTP and LTD) while the postsynaptic modifications are not (and express only LTP), at least on the timescale of whole-cell recording experiments. We have now clarified this further. Interestingly, there is some evidence for such a reversal in the cerebellum and different locus expression pattern in the hippocampus (see Discussion).

*I am not disputing the premise, however. It may well be that the slower turnover of presynaptic terminals compared to spines allows faster relearning. I believe that Finnerty and colleagues have done some work on this and they should probably be mentioned in this paper*.

We have now cited Cheetham et al. (Cereb Cortex 2014) and Finnerty et al. (Nature 1999) in the Discussion, mentioning that changes in experience can affect both pre- and postsynaptic structures, which appear to be consistent with our model, although perhaps with different time constants.

*The second point is that the main body of the paper seems to be over very quickly. I was left searching for more text. There are several points that could be more fully discussed, including the elements that are explicitly missing from the model and what the impact might be. For example, how does inhibition and plasticity of inhibition impact the outcome and how might the time course of structural plasticity modify the model. On the second point, the Matsuzaki reference deals with structure and synaptic weight post-synaptically but not pre-synaptically*.

The [23] paper only looks at postsynaptic structure, but their results are consistent with our predictions (i.e. that the postsynapse should outlast input changes). Given the Matsuzaki results, our results might explain their results on the postsynaptic side. Our model further predicts that the presynapse should gate the input, which might be reflected in the structure of the presynapse. However, to the best of our knowledge, this has not been tested before. We have added a reference to the review by Holtmaat, Nature Reviews Neuroscience, 2009.

*It would also be useful to hear how the model might fit different structures - would more presynaptic plasticity be warranted in layer 5 pyramidal cells than in hippocampus for example?*

Thank you for pointing this out. We have added a discussion on this. Briefly, in analogy with the triplet model results from [39], we argue that our modeling framework should apply to those brain regions as well. However, model parameters will be different, the consequences of which will be difficult to know without detailed studies.

Reviewer #3:

*It is very interesting that the rule results in better discrimination. What is not clear is why? That is, why does a higher P value provide improve SNR? Yes, the first presynaptic spike generates a large reliable EPSP, but the subsequent spikes are less reliable and depressed. The SNR and ROC analyses were analytical, but I'm struggling to understand how these analyses can capture the true SNR and discriminability without taking into account the firing rate of the presynaptic units. Discriminability must depend in part of on the frequency of the Poisson inputs (and the D and F time constants)-but as it stands the analysis is based on the probabilistic nature of P, that is, essentially only for the first spike. Either I'm missing something, or an analysis based on the actual simulations (which take into account the timing of the spikes) should be performed. This is probably a minor issue, however, as*Figure 3*makes it clear the discriminability based on firing rate is excellent*.

*Based on*Figure 3*, P seems to saturate at 1, meaning that there is little or no release for the next 200 ms. This is one reason the SNR calculation may be misleading. It would be useful to show postsynaptic trace segments during stimulation as part of this figure*.

We thank the reviewer for highlighting this. We have now clarified that the SNR and discrimination analysis was done for the first EPSP alone. A full characterization of the information transmission requires numerous assumptions about the neural code and the input statistics. Even so, we have now included a supplement to Figure 2—figure supplement 3, showing how SNR and information transmission is improved by our learning rule when short-term plasticity is considered.

*I think some readers, particularly, those accustomed to the hippocampal field, will be unfamiliar with the strong evidence that there are presynaptic components to STDP. So I think it is important to strengthen their argument a bit, and perhaps mention that, these studies are based on neocortical data, and that there is likely a difference between hippocampal and neocortical STDP*.

We have now clarified this in the Discussion.